# Areas of Research Focus and Trends in the Research on the Application of VR in Rehabilitation Medicine

**DOI:** 10.3390/healthcare11142056

**Published:** 2023-07-18

**Authors:** Chen Wang, Jingqi Kong, Huiying Qi

**Affiliations:** 1Department of Health Informatics and Management, The School of Health Humanities, Peking University, Beijing 100191, China; wangchenparis@bjmu.edu.cn; 2Department of Language and Culture in Medicine, The School of Health Humanities, Peking University, Beijing 100191, China; 18811531265@pku.edu.cn

**Keywords:** VR, rehabilitation, bibliometric, research focuses, development trend

## Abstract

Objective: To establish the areas of research focus in the application of VR in rehabilitation medicine, analyze its themes and trends, and offer a reference for future related research in this field. Methods: This paper provides an in-depth analysis of the development process, areas of research focus, and research trends in the field of the application of VR in rehabilitation medicine, using the Web of Science core dataset as the source and using a bibliometric analysis with CiteSpace. Results: The application of VR in rehabilitation medicine was composed of three stages, and the research topics were reviewed from five perspectives: neurological rehabilitation, psychological treatment, pain distraction, cardiopulmonary rehabilitation, and visual–spatial disorder. Limitations: The research data were sourced from the Web of Science core dataset only, and the data-sample size was not comprehensive. Conclusions: Overcoming VR-technology-induced vertigo, mental disorders from the overuse of VR, individualized treatments, and integration with traditional therapy are all challenges in the application of VR in rehabilitation medicine that require research. In addition, developing VR products with better experiences, constructing standardized guidelines, and conducting more high-quality clinical studies are all future research topics related to the application of VR in rehabilitation medicine.

## 1. Introduction

Virtual reality (VR) is a multidisciplinary technology that generates a virtual digital world through computer simulations and provides complex, realistic, and immediate feedback similar to reality using software programs and hardware devices (e.g., headphones, helmets, glasses, gloves, vests, etc.) to give users an immersive feeling and experience [1]. In recent years, VR technology has gradually entered and been applied to the medical field, especially in rehabilitation.

Rehabilitation medicine aims to improve the quality of life of sick, injured, and disabled people and their eventual integration into society. Because VR has the characteristics of virtual tasks, interesting scenarios, and high levels of user participation, VR technology can be a realistic and convenient tool for all rehabilitation subjects to adapt to the real world in advance, and it can achieve excellent therapeutic effects in rehabilitation. In recent years, VR technology, as a frontier technology in rehabilitation medicine, has shown great advantages and is increasingly used in multiple fields, such as neurological rehabilitation, pediatric rehabilitation, orthopedic rehabilitation, etc. For example, the use of a commercial wearable head-mounted display (HMD) and selected VR motion games for Parkinson’s disease to maintain physical and functional abilities has shown good disease-management effects [2]; a VR-based interactive cognitive training (VICT) system shows good effects on the cognitive rehabilitation of traumatic brain injury (TBI) in children [3]; and VR technology can also be combined with existing rehabilitation methods, such as walking exercises combined with virtual reality, which can improve gait and balance problems significantly [4].

After reviewing relevant studies, this research found that although there are numerous studies on this topic, most of them focus on the feasibility and effectiveness of VR technology for rehabilitation training for certain types of patients. To the best of our knowledge, there has been no systematic research on the areas of research focus and research trends in the application of VR in rehabilitation medicine. Therefore, this paper systematically reviews the literature on the application of VR in rehabilitation medicine and summarizes and analyzes the areas of research focus, thematic evolution, and development trends in this field through bibliometric methods, aiming to deepen the understanding of the application of VR in rehabilitation medicine and provide a reference and theoretical basis for subsequent related research.

## 2. Study Design

### 2.1. Data Acquisition

This article uses CiteSpace as a bibliometric analysis tool. Currently, Citespace software allows the individual analysis of research imported from various databases, such as PubMed, Web of Science, Scopus, etc. However, it does not currently offer the ability to merge and analyze research from multiple databases together. Among the various databases, the Web of Science Core Collection is a comprehensive academic database that covers multiple disciplines. The research indexed in the Web of Science Core Collection holds high academic value and reliability. Hence, it was chosen as the data source for this study.

### 2.2. Search Strategy

The search strategy was as follows: TOPIC: (“virtual reality*” OR “VR”) AND (“occupational therapy*” OR “physical therapy*” OR “physical medicine” OR “rehabilitation” OR “physicotherapeutics”). The item type was “article”, and the time range was up to 2022. In total, 3448 articles were obtained. The retrieval date was 16 January 2023.

### 2.3. Methodology

CiteSpace is a literature information visualization tool developed by Prof. Chaomei Chen’s team at Drexel University, USA. This study first used Excel 2019 to conduct a basic statistical analysis of the number of annual publications; then, national (regional) collaboration network analysis of the literature was conducted through CiteSpace, as well as keyword co-occurrence network and co-word timeline network analysis to explore the development history, areas of research focus and frontier topics of VR in rehabilitation medicine research [5].

## 3. Data Analysis

### 3.1. Temporal Distribution Analysis

The interannual variation in the number of publications can visually reflect the distribution of the literature on the application of VR in rehabilitation medicine. Figure 1 shows the temporal distribution of the research literature on the application of VR in rehabilitation medicine, the earliest of which was the paper about rehabilitation for bilateral vestibular hypofunction in 1993 [6]. Before 2002, there were few studies related to the application of VR research in rehabilitation medicine; from 2003 to 2012, the number of related articles in this field showed a continuous upwards trend; and especially from 2013, it gained a faster growth rate, which indicated that research on the application of VR in rehabilitation medicine is attracting increasing attention.

### 3.2. Spatial Distribution Analysis

By mapping national (regional) collaborative research, it is possible to visually observe the sources of academic productivity of research on the application of VR in rehabilitation medicine and to understand the core strengths and mutual collaborative relationships in this research field. In this study, CiteSpace was used to generate a national collaborative network, as shown in Figure 2. Every node represents a nation (region), the node size represents the total number of published articles, and the different colors in the node correspond to the number of articles in different periods. The color bar at the top of the figure indicates the year changes. The earliest is to the left, and the more to the right one looks, the newer the year. Thus, the year of the center of the circle is the earliest. The edge between nodes stands for national collaboration, and the different times of starting cooperation are distinguished by the different colors of the edge. It can be inferred that the United States takes a core position and has collaborated with several countries, and started earlier than most other countries and regions.

Table 1 presents information about the top ten countries/regions in terms of the number of publications. The United States topped the list with 1088 articles, accounting for more than 31% of the total, which demonstrates the importance that the United States attaches to this research field. Italy, Canada, Spain, China, and South Korea have all published more than 200 articles. The top 10 countries and regions accounted for more than 84% of the total number of articles published, indicating that the main research strength on the application of VR in rehabilitation medicine is concentrated in these countries and regions. Centrality is adopted to measure the importance of nodes in a network and is a key metric of interest in the fields of data visualization and scientometrics. Therefore, in CiteSpace, the centrality of each node (e.g., countries, keywords, and literature) is calculated to discover and assess its importance. This is achieved by highlighting nodes with high centrality using purple circles. With regard to centrality, the United States ranked first with 0.36, followed by Spain and Australia, ranked second and third with 0.21 and 0.2, respectively. These country/region nodes are marked with purple circles to indicate the importance of the node. The United States started publishing relevant articles in 1997; Italy (2000), England (2001), Canada (2003), and South Korea (2003) commenced relevant studies in approximately 2002, and other countries/regions started publishing relatively later and began publishing after 2007. It is evident that the United States has a comparative advantage in terms of both the number of articles and their significant impact.

### 3.3. Keywords Statistical Analysis

By analyzing the keyword co-word network of the literature, the relevance of keywords to the research focus in specific academic fields can be explored. Figure 3 shows the co-word network of keywords generated by CiteSpace. Every node stands for a keyword. The larger the node is, the higher the occurrence frequency of the keyword. The color bar at the top of the figure represents the year changes. The earliest is on the left, and the more to the right one looks, the newer the year. Additionally, the different colors in the node describe the occurrence frequency of the keyword in different periods; thus, the year of the center of the circle is the earliest. The node with high centrality is marked with a purple outer ring to present the importance of the keyword. Among them, the significant nodes with high frequency and centrality include virtual reality (1924 times, 0.16), gait (306 times, 0.1), video games (281 times, 0.14), and environment (233 times, 0.15), all of which are key issues in this research field. In addition, keywords with high word frequency include rehabilitation (1666 times), stroke (1034 times), exercise (537 times), balance (360 times), children (208 times), older adults (173 times), cerebral palsy (159 times), telerehabilitation (115 times), and Parkinson’s disease (109 times), which offer a reference for the following review of research themes.

### 3.4. Topics Evolution Analysis

Figure 4 demonstrates the development of keywords in each cluster with the purpose of promoting the understanding of the theme evolution path of this research field. The position of the keyword node on the timeline shows the year when the keyword first appeared, and the keyword cluster label generated by CiteSpace is displayed on the right. From the frequency of keywords appearing in the research themes, it can be seen that the main research focus is on stroke, cognitive rehabilitation, depression, and cerebral palsy. It can be found that the research themes of VR in rehabilitation show diverse development, but it always revolves around the extension of the application of VR in rehabilitation research for different diseases. Hence, this study divides its development into three stages.

#### 3.4.1. Exploratory Stage (Since 1995)

With the emergence of VR products, researchers began to discuss the existing and potential applications of VR technology in rehabilitation, such as the prospect of the application of VR in neurological rehabilitation [7] and the exploration of its efficacy for motor rehabilitation and phobias [8]. Meanwhile, VR technology has started to be applied to develop products for rehabilitation, such as VR rehabilitation training employing a diagnostic glove and WorldToolKit graphics library [9] and the development of a bedside rehabilitation system, which allows patients to experience wandering through a virtual forest, aiming to improve the quality of life of bedridden patients and the elderly [10].

Furthermore, VR-based therapies have commenced exploration of their application in a variety of diseases, such as the visual and auditory stimulation generated by VR technology for communication with and the education of autistic children [11]. Additionally, another representative example is the VR exposure therapy adopted by researchers at Emory University to simulate a Vietnam war zone for the treatment of PTSD disorders in veterans, which has been proven to be effective and is still used in the medical field today [12].

#### 3.4.2. Developmental Stage (Since 2010)

In 2010, Microsoft launched the motion-tracking sensor “Kinect”, which has made significant advances in both graphics processing and motion capture technology, further promoting research on the application of VR in rehabilitation medicine. Designing rehabilitation games through Microsoft Kinect for sports rehabilitation exercises for Parkinson’s disease (PD) patients could effectively motivate their exercise interest [13]. Hemiplegic stroke survivors trained with Xbox Kinect could significantly improve their multiple aspects of upper limb function, such as range of motion, motor function, and overall hand dexterity [14]. The active virtual reality gaming (AVG) of Xbox 3 Kinect made motor skill practice available and contributed to the rehabilitation of children with developmental coordination disorder (DCD) [15].

Another representative product is Oculus Rift, an immersive virtual environment using a very inexpensive wide field of view, which was launched by Oculus in 2013. Oculus Rift Virtual Reality (VR) goggles could elicit a strong illusion of presence and reduce pain through VR, which can be applied to daily burn wound care and physical therapy, enabling more patients (possibly even at home) to conveniently use VR for pain control [16]. Analysis of data collected by VR hardware can provide deep insight into neck flexibility and overall neck motion during head movement, which is applicable to assess pain-related stiffness as well as monitor progress [17].

In 2014, Google released its VR product Cardboard. Because of its low price and positive experience, its application in rehabilitation has also developed rapidly, which was beneficial for the rehabilitation therapy of patients with mechanical strabismus and/or amblyopia who have lost eye movements [18]. An immersive experience based on Google Cardboard allows for long training that can effectively reduce pain intensity and is a powerful intervention to enhance motor function and reduce pain perception [19].

#### 3.4.3. Promotion Application Stage (Since 2019)

At this stage, new VR products are constantly being launched, represented by Meta Quest 1 released in 2019 (Meta Quest 2 released in 2021), more commonly known as Oculus Quest. Meta Quest is a standalone device with high degrees of freedom and hand-tracking capabilities. Meta Quest offers various applications in the field of rehabilitation, such as shoulder rehabilitation [20], distal radius fracture rehabilitation [21], therapy in lateral epicondylitis patients [22], knee osteoarthritis patients [23], walking activities in pediatric neurorehabilitation [24], and chronic low back pain [25,26].

Additionally, as VR technology matures, a growing number of manufacturers have developed personalized VR devices for different patients to build targeted training scenarios. The serious game REHAB FUN VR environment developed with the Unity engine helped children with cerebral palsy improve their motor and cognitive abilities [27]. A novel, 3D, customizable, personalized home exergaming for Rehabilitation system was introduced to rehabilitation training for stroke patients [28]. The next-generation BrightArm Compact (BAC) rehabilitation table and its novel game controllers were designed for ambidextrous training, forward/backward rotation, grip strengthening, and finger extension [29]. Meta Quest and a motion tracking system were used for upper limb rehabilitation [20]. The application of VR in rehabilitation was found to be effective in incentivizing patients and maintaining a high level of patient compliance, matched with the abundant training scenarios of the reward and punishment mechanism [30]. In addition to providing adequate scenarios, the application of VR technology also enhanced the potential for neuroplasticity in the rehabilitation process [31].

## 4. Analysis of Research Topics on the Application of VR in Rehabilitation Medicine

From the analyses of keyword clustering and theme evolution, the research on the application of VR in rehabilitation medicine involves gait balance, stroke, Parkinson’s disease, cerebral palsy, neurorehabilitation, cognitive rehabilitation, brain injury, pulmonary rehabilitation, depression, etc. Meanwhile, 3448 papers were selected for abstract reading by the co-authors of this study and classified by research topic based on abstracts. Eventually, based on high-frequency keyword statistics, clustering, and theme evolution path, and combined with the content of the classical literature by co-authors, the research themes are summarized into the following five dimensions for review, using highly cited literature from diverse classifications as representative examples (as shown in Figure 5).

### 4.1. Neurological Rehabilitation

Neurological rehabilitation mainly focuses on the rehabilitation of dysfunctions caused by neurological diseases, such as motor, sensory, verbal, deglutitive, and cognitive disorders, involving conditions such as stroke, cognitive disorders, Parkinson’s disease, traumatic brain injury, vestibular rehabilitation, and neurological rehabilitation in children, as shown in Table 2. VR technology has incomparable advantages in neurological rehabilitation, and neurological rehabilitation is one of the directions where VR attains most applications in rehabilitation.

#### 4.1.1. Stroke Rehabilitation

Stroke is one of the most serious neurological diseases. The immersive experience and interactivity of VR can increase the efficiency of stroke rehabilitation training and motivate patients’ cooperation.

In terms of improving cognitive function in stroke patients, immersive VR environments can improve the psychological state of post-stroke patients, especially anxiety symptoms [32,33], and can effectively reduce the time patients spend in the hospital [34].

With respect to recovering limb function, studies have shown that the addition of VR technology to conventional physiotherapy (CP) procedures resulted in significant improvements in balance and motor function over CP-treated stroke patients [35]; and combining VR therapy with mirror therapy (MT) exercises had the potential to replace traditional physiotherapy in the rehabilitation of lower limbs after stroke [36].

In addition, virtual reality is considered to be an efficient adjunctive therapy for the functional recovery of verbal ability in post-stroke aphasic patients [37], and the application of VR specialized in verbal rehabilitation in aphasic patients has been developed [38].

#### 4.1.2. Geriatric Cognitive Impairment

Mild cognitive impairment (MCI), also referred to as the predementia stage, is one of the most common cognitive dysfunctions seen in the elderly population. Studies have suggested that virtual reality-based cognitive training (VRCT) can promote brain, cognitive, and physical health in older adults with MCI [39]. VR-based cognitive-motor rehabilitation (VRCMR) can lead to higher subject interest and motivation for training with the purpose of increasing training participation [40], bringing about greater improvement in the areas of visual–spatial perception, visuomotor organization, orientation, thinking operation, and attention than the conventional cognitive rehabilitation (CR) intervention group [41].

#### 4.1.3. Parkinson’s Disease

VR has also been proven to be effective in the cognitive and motor rehabilitation of Parkinson’s patients. The corresponding VR cognitive application or training system on the cognitive and behavioral recovery of Parkinson’s patients was able to exercise their memory, executive functions, attention, logical thinking, and speed of thought, thus strengthening their cognitive functions in terms of executive and visuospatial abilities [42,43]. Furthermore, patients with Parkinson’s disease showed performance improvements in learning or retention after Nintendo Wii Fit™ training [44].

The effect of VR on the motor rehabilitation of Parkinson’s patients is also evident. Virtual reality and motion picture training combined with conventional physiotherapy can remarkably improve resting tremors, rigidity, posture, gait, and slow body movements in Parkinson’s patients [45]. A commercial wearable head-mounted display (HMD) matched with the VR motion game, a boxing exergame, is practical for rehabilitation work in a variety of settings for mild to moderate Parkinson’s patients [2]. Nonimmersive virtual reality exercise games can enable older Parkinson’s patients to effectively improve their gait and balance [46], and can be conducive to motivating patients’ confidence and motility [47].

#### 4.1.4. Traumatic Brain Injuries

Traumatic brain injury (TBI) is a devastating injury that can lead to deficits in multiple domains. Virtual reality combined with treadmill training is safe and feasible for TBI patients. Those who accepted VR balance interventions presented greater enjoyment of training compared to those who participated in conventional therapeutic approaches and those who just used a treadmill [48]. Meanwhile, the combination of VR training led to greater improvement in patients’ cognitive performance [49].

#### 4.1.5. Vestibular Rehabilitation

Motion sickness (MS) can be triggered by direct or indirect stimuli due to mismatches in the visual–vestibular autonomic pathways. VR proved to be significantly effective and beneficial for MS rehabilitation. According to the study, the regular and frequent application of visual stimulation in VR games for MS patients has a positive impact on vestibular adaptation mechanisms [50], improving dizziness, balance, gait, effects of fatigue, quality of life, and muscle tone [51]. It is evident that immersive VR vestibular training programs provide a more enjoyable rehabilitation manner [52], increasing the efficiency of the process and reducing the risk of inadaptability to exercises.

#### 4.1.6. Neurological Rehabilitation in Children

Virtual reality technology offers immersive training content that is both appealing to children and effective in some neurological disorders in children, such as cerebral palsy and children’s dyslexia. The study demonstrated the feasibility, safety, and initial efficacy of VR-based interactive cognitive training (VICT) systems for executive function rehabilitation in TBI children [3]. VR-based rehabilitation enhanced hemiplegic cerebral palsy (HCP) children’s cognitive functions, such as orientation, spatial perception, praxis visuomotor construction, and thinking operations [53], and considerably improved the reaction time of cerebral palsy children [54]. The Horse-Riding Simulator (HRS) with VR is a complementary therapy for rehabilitation in spastic cerebral palsy children, which showed an excellent effect for rehabilitation in spastic cerebral palsy (CP) children in terms of overall motor function and balance control [55]. Moreover, the virtual reality rehabilitation system (VRRS) can prolong the training time and improve the treatment compliance of children with dyslexia [56].

### 4.2. Psychological Treatment

VR creates a secure environment for patients to experience and explore freely. Patients’ various perceptual activities, such as vision, hearing, and touch, as well as emotional reactions, such as joy, sadness, tension, and fear, can be fully expressed. These technical features of VR are well suited for psychological treatment (Table 3).

In autism spectrum disorder treatment, VR is a proper and promising tool to improve the cognitive function of individuals who are severely affected by autism spectrum disorder [57]. The utilization of virtual reality technology can intervene and encourage cognition, imitation, and social interaction in children with autism spectrum disorders, and VR-based rehabilitation contributes to the cognitive-developmental abilities and social communication skills of ASD children [58].

In depression treatment, VR therapy integrated with neurological rehabilitation has a positive effect on improving mood and reducing depressive symptoms in post-stroke patients [59], and a feasible plan for the treatment approach has been published [60], which is considered to be a helpful nonpharmacological intervention for post-stroke depression [61].

In schizophrenia treatment, multimodal adaptive social interventions in virtual reality are feasible and acceptable in improving social functioning and clinical outcomes in schizophrenia patients [62]. For example, a cinematic VR-based application, cinematic VR, has shown satisfying acceptability in schizophrenia patients [63]. A VR-based vocational rehabilitation training program developed for schizophrenic patients can strengthen their general psychosocial functioning and memory [64].

### 4.3. Pain Distraction

VR has been applied for pain distraction in a wide range of known painful medical procedures. For example, Oculus Rift VR goggles can assist in pain control in the occupational therapy of pediatric burn patients and can be employed for pain distraction in the rehabilitation of burnt children [16]. Additionally, VR can be used as a powerful nonpharmacological pain reduction technique for adult burn patients during physical therapy and potentially for other pain procedures or pain populations [65]. For young children with burns, the employment of VR during physical therapy is more enjoyable and may allow them to experience less anxiety [66].

### 4.4. Cardiopulmonary Rehabilitation

Medical services in cardiopulmonary rehabilitation have been successfully carried out through virtual reality technology, and the specific applications are shown in Table 4.

Regarding cardiovascular rehabilitation, VR, as an assistant tool to conventional cardiac rehabilitation (CR) programs, performs well in promoting compliance and satisfaction in stage II ischemic heart disease patients compared to the conventional treatment group [67]. The use of exercise games adding virtual reality therapy in cardiac rehabilitation patients led to higher scores in heart rate, respiratory rate, and perceived exertion scores during their period of performance and 5 min after treatment [68].

In lung rehabilitation, an effective improvement can be made in terms of lung function, cognitive function, exercise tolerance, and medication efficiency through VR-based pulmonary rehabilitation training, which can also reduce the symptoms of dyspnea in elderly chronic obstructive pulmonary disease (COPD) patients with mild cognitive dysfunction (MCI) [69].

Rehabilitation was a critical aspect of healthcare systems during the COVID-19 pandemic, and COVID-19 survivors suffered from reduced lung function, critical illness polyneuropathy and myopathy, and cardiorespiratory deconditioning. Therefore, it is important to prepare new interventions that allow healthcare providers to maximize responses to rehabilitation challenges. VR can provide a differentiated form of intervention during hospitalization for COVID-19, and virtual reality software can have a positive impact on the motivation and engagement of COVID-19 patients compared to conventional physical therapy interventions. Apart from in-hospital treatment, cognitive rehabilitation can also be delivered through a remote rehabilitation modality in conjunction with VR [70]. The study implied that VR-based rehabilitation is good for exercise tolerance and reduces anxiety and depressive symptoms in COVID-19 patients [71].

### 4.5. Spatial Neglect

VR also shows good performance in improving patients’ spatial neglect (SN) and is regarded as an effective way to promote visual field recovery, spatial cognition, and mood in patients with unilateral spatial neglect after stroke [72]. Satisfactory therapeutic effects were obtained by using virtual reality in 3D space to examine and treat unilateral spatial neglect [73]. In addition, the immersive and modifiable aspect of VR to translate Musical Neglect Therapy (MNT) into a VR therapy tool can increase patients’ enjoyment of the treatment [74].

## 5. Challenges and Future Trends

The rapid progress of VR technology has introduced a new vision for clinical rehabilitation treatment by providing a new means of rehabilitation, higher treatment compliance, and better patient management in a hospital or home environment, with better efficacy for a wide range of diseases. The application of VR technology will be an important direction in the future development of rehabilitation treatment. At present, some achievements have been attained in the research on VR technology in rehabilitation, but there are still many issues to be solved by rehabilitation experts and engineering experts. Hence, this study presents the following challenges and future perspectives, taking into account the current research focus and topic evolution development.

### 5.1. Current Challenges

#### 5.1.1. VR-Induced Motion Sickness Disorder

Current VR technology mainly depends on large projection screens or head-mounted displays to generate VR environments. Due to the inconsistency between visual and physical perceived motion, visually induced motion sickness disorder may occur, with symptoms such as headache, sweating, nausea, vomiting, fatigue, and disorientation, and may also cause safety issues and other health problems [75,76]. Motion sickness disorder is the main obstacle to overcome for the current immersive VR technology.

#### 5.1.2. Mental Disorders Caused by VR Dependence 

A fully immersive VR environment causes the juxtaposition of perception and hallucination, blurring the boundary between reality and hallucination [77,78]. When users are keen on the virtual environment and interpersonal interactions, are overly dependent on VR companions, or their nerves are connected to the VR world, they may lose themselves, ignore real-world problems and real interpersonal relationships, and then suffer mental disorders. In this sense, VR is no different from electronic drugs [79]. Therefore, attention should be paid to reasonable training schedules and training content.

#### 5.1.3. Individualized Treatment

Rehabilitation often requires personalized treatment plans tailored to each patient’s specific condition and progress. VR applications need to be flexible and adaptable to cater to the diverse needs of patients [80]. Customization options, such as adjusting difficulty levels or targeting specific motor skills, should be available to ensure maximal effectiveness.

#### 5.1.4. Integration with Traditional Therapy

Integrating VR technology into traditional rehabilitation practices can be a complex process. Rehabilitation experts and engineering experts must collaborate to ensure seamless integration of VR technology with existing treatment approaches [81]. This includes developing standardized protocols, defining best practices, and establishing guidelines for combining VR with other therapeutic modalities.

### 5.2. Future Prospects

#### 5.2.1. Improving User Experience

At this stage, the sensitivity of most sensors cannot fully meet the rehabilitation needs of patients. It is expected that researchers can develop VR products offering a full range of somatosenses as soon as possible that can give users a full-body experience containing the senses of touch, hearing, smell, and taste. Meanwhile, researchers need to pay attention to how to reduce the discomfort of users’ experience, increase their emotional experience and adjust their emotions, allowing the virtual environment and the real environment to achieve a seamless connection. In addition, VR rehabilitation products also aim to create a virtual entity that integrates initiative, responsiveness, autonomy, sociality, mobility, and adaptability to meet the diversified needs of patients in the aspects of movement, speech, emotion, and social interaction.

#### 5.2.2. Develop Standardized Guidelines

Currently, VR technology’s application in rehabilitation lacks more standardized treatment guidelines to guide the development of VR rehabilitation programs. More research, especially randomized controlled trials, is still needed in the future to confirm the effectiveness of VR rehabilitation programs.

#### 5.2.3. Conducting More High-Quality Clinical Studies

Owing to the high cost of VR equipment, the existing clinical application studies generally have a small sample size of subjects. Moreover, there is variability in the effects of patients applying VR technology; thus, the rehabilitation training conditions are unclear. The effectiveness of its clinical application is still expected to be found in more high-quality studies.

## 6. Conclusions

This article presents a bibliometric analysis of research on the application of VR in rehabilitation medicine. The main findings are represented in the following aspects.

Through the analysis of the temporal distribution graph, this article discovers trends in the research output of the application of VR in rehabilitation medicine. From the initial exploration of how to apply virtual reality in rehabilitation, research on the application of VR in rehabilitation medicine has continued for 30 years and has received growing attention from researchers. Especially after 2013, the number of studies has grown dramatically, and numerous research results have been achieved.

The distribution of this research field in the world was confirmed through the analysis of a national (regional) collaboration network. The United States, Italy, Canada, Spain, and China are the major countries for research on the application of VR in rehabilitation medicine, while the United States exceeds other countries in terms of both the number and influence of the literature. Globally, the proportion of multinational and cross-regional collaborations in the research on the application of VR in rehabilitation medicine is increasing, and the research is gradually becoming more specialized, comprehensive, and holistic.

Through a keyword co-word timeline network analysis, we find the core literature and topic evolution pathways of the research on the application of VR in rehabilitation medicine. This study divides the research on the application of VR in rehabilitation medicine into three stages: the exploratory stage, the developmental stage, and the promotion application stage. The research content of each stage is related to the technological development at that time.

A thematic review of the research on the application of VR in rehabilitation medicine was summarized by analyzing the keyword co-word network. The areas of research focus on the application of VR in rehabilitation medicine are diverse. The research themes in several areas of neurological rehabilitation, psychological treatment, pain distraction, cardiopulmonary rehabilitation, and visual–spatial disorders all reflect the main focuses of research on the application of VR in rehabilitation medicine.

Finally, this article presents the current challenges and future prospects for the application of VR in rehabilitation medicine. Overcoming VR-induced motion sickness disorder and mental disorders caused by excessive VR use are both challenges for this research field. Furthermore, developing VR products with better experience, constructing standardized guidelines, and conducting more high-quality clinical studies are all future research trends for the application of VR in rehabilitation medicine.

This research systematically sorts and summarizes the relevant research on the application of VR in rehabilitation medicine and provides references for future practice and exploration in this field. However, this study also has limitations. Only the literature in the WOS core dataset was searched and analyzed; thus, the data sample was not comprehensive. Further studies will expand the scope of the literature search for a more comprehensive analysis.

## Figures and Tables

**Figure 1 healthcare-11-02056-f001:**
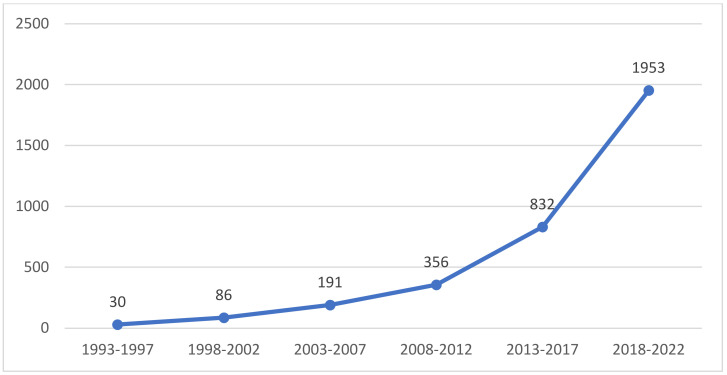
Temporal distribution graph of research on the application of VR in rehabilitation medicine.

**Figure 2 healthcare-11-02056-f002:**
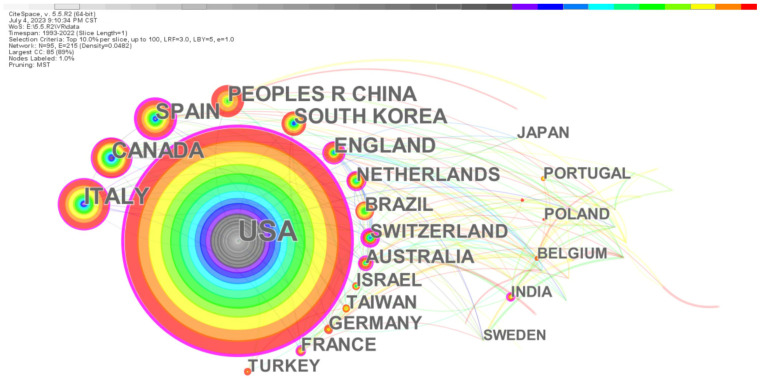
National (regional) collaborative network of research on the application of VR in rehabilitation medicine.

**Figure 3 healthcare-11-02056-f003:**
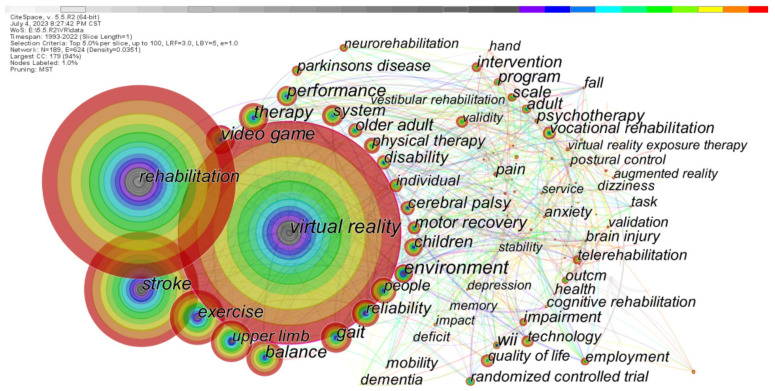
Co-word network of keywords on the application of VR in rehabilitation medicine research.

**Figure 4 healthcare-11-02056-f004:**
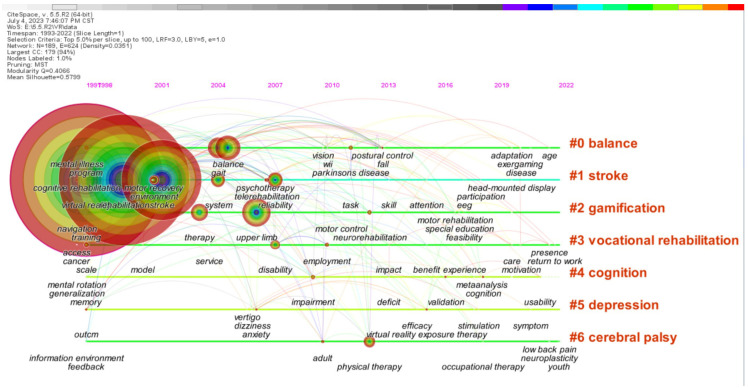
Co-word timeline view of research literature on the application of VR in rehabilitation medicine.

**Figure 5 healthcare-11-02056-f005:**
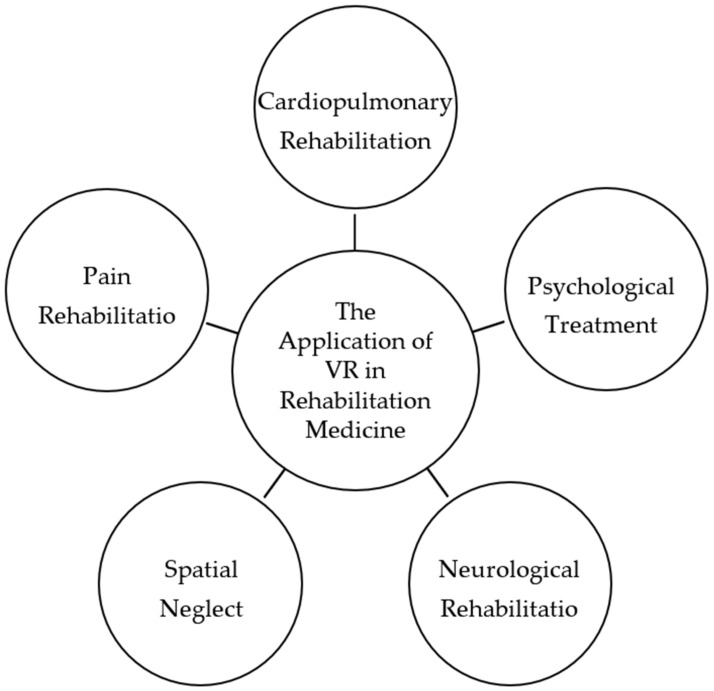
Research topics on the application of VR in rehabilitation medicine.

**Table 1 healthcare-11-02056-t001:** Country/region and number of published articles.

Country/Region	No. of Articles	Percentage	Frequency of Citations	Centrality	Year of First Publication
USA	1088	31.555%	1010	0.36	1997
Italy	333	9.658%	262	0.17	2000
Canada	235	6.816%	205	0.16	2003
Spain	234	6.787%	185	0.21	2007
People’s R. China	209	6.061%	162	0.09	2012
South Korea	200	5.800%	155	0.09	2003
England	193	5.597%	148	0.1	2001
Australia	136	3.944%	97	0.2	2007
Brazil	135	3.915%	103	0.06	2013
Netherlands	135	3.915%	104	0.16	2008

**Table 2 healthcare-11-02056-t002:** Research on the application of VR in neurological rehabilitation.

Type of Disease	Conclusions	Literature
Stroke	The immersive experience and interactivity of VR can stimulate patients’ active cooperation and improve patients’ cognitive and psychological states.	[32,33,34]
	VR technology can help stimulate stroke patients to recover limb function.	[35,36]
	Conversational therapy in the VR environment is useful for language recovery in post-stroke aphasia patients.	[37,38]
Geriatric cognitive impairment	Virtual reality-based cognitive training (VRCT) enhances the brain, cognitive, and physical health of older adults with MCI, and it is more effective than traditional therapies.	[39,40,41]
Parkinson’s disease	VR improves the cognitive function, learning, and memory of Parkinson’s patients.	[42,43,44]
	VR has a significant effect on motor rehabilitation in Parkinson’s patients.	[45,46,47]
Traumatic brain injury	VR boosts cognitive and behavioral abilities in traumatic brain injury (TBI) patients.	[48,49]
Vestibular rehabilitation	Regular and frequent application of visual stimulation has a positive influence on the vestibular adaptation mechanism.	[50,51,52]
Neurological rehabilitation in children	Rehabilitation intervention based on VR improves the cognitive function of hemiplegic cerebral palsy (HCP) children.	[3,53,54,55]
	VR intervention for children with dyslexia promotes treatment outcomes.	[56]

**Table 3 healthcare-11-02056-t003:** Research on the application of VR in psychological treatment.

Type of Disease	Conclusions	Literature
Autism spectrum disorder	VR technology for rehabilitation benefits the cognitive development and social communication skills of ASD children.	[57,58]
Depression	Immersive VR therapy is an effective nonpharmacological intervention for post-stroke depression.	[59,60,61]
Schizophrenia	Multimodal adaptive social intervention with VR (MASI-VR) offers a recovery pathway for schizophrenia patients.	[62,63,64]

**Table 4 healthcare-11-02056-t004:** Research on the application of VR in cardiopulmonary rehabilitation.

Type of Disease	Conclusions	Literature
Cardiovascular rehabilitation	Virtual reality, as a supplementary tool of traditional cardiac rehabilitation (CR) programs, plays a positive role in treatment.	[67,68]
Lung rehabilitation	VR rehabilitation therapy can improve patients’ lung function, cognitive function, and exercise tolerance.	[69]
COVID-19 treatment	VR intervention offers a differentiated form of intervention that positively affects people being treated for COVID-19.	[70,71]

## Data Availability

Not applicable.

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
