# Peer review of "Areas of Research Focus and Trends in the Research on the Application of VR in Rehabilitation Medicine"

_healthcare, 2023, doi:10.3390/healthcare11142056_

Round 1

Reviewer 1 Report

This paper gives an overview about recent trends in VR applications regarding rehabilitation. I have found the paper interesting and worth for publishing as a review paper. I have some concerns and questions that could improve readability and understanding, listed as follows. 

My recommendation is acceptance with revision.

Abstract: why only did you use WoS for your research, if you say yourself it is not comprehensive? It is a good one, but there also other data bases - this kind of papers usually include other data bases as well.

Writing: sometimes a _space_ is missing between words and reference (e.g.,[1], [4]...).

Line 59: What is Bradford's law? You say, "according to" it, but I think it was about the amount and trends of references in journals, not about the fact that you have to search for good papers in good journals (which is true, but I don't see what Bradford's law has to do with it). Also a reference may do good to support your statement or choice for your work.

For TOPICs, you also included AR and MR? You should mention this in the introduction, because AR is not VR, especially in case of vertigo and other symptoms associated with VR. The title also suggest "VR only", although including AR is different. Furthermore, it would be good to explain this (differences between them) in the intro, followed by your decision to include them (or not). Later in the paper, I did not find AR solutions, so there is a confusion. E.g., is Kinect part of VR? I assumed HMDs are included only. 

You have asterisk (*) here, but I could not find what this means.

I know, there are unlimited possibilities for keywords, but I was wondering why "telerehabilitation" or "telemedicine" is not included? That would have been my first to search for (which I had done indeed as we prepared a paper for AR/VR telerehab for stroke patients).

I like figures 1 and 2, but figure 2 needs more explanation, what to see. What are the colors? And what is the size of the circles representing? There are also different colored lines linking the circles. Some readers (like me) may be not very familiar with this graphical representation. The figure legend should explain more what to see. 

The same is true for figure 3, where we also have overlapping circles. These figures are good to explore and visualize, but readers need more help to use them.

Lines 138-139 are repeated in 141-142.

Section 3.4.3.

Are there any results about/with Meta Quest 1 and/or 2? For the keyword search "oculus quest" Google Scholar gives 10.000+ results and with "on balance" extension still 3800 (after 2019). 

I found sections 4 and 5 really good. 

To section 4: I know, it is impossible to find and evaluate _all_ related articles, even in a limited amount (e.g., those that are listed in a database). You said you found 2940 articles. Here, you referenced some of them [27-67]. How did you select these for deeper evaluation? What was the criteria to include these? It is OK to highlight some important or representative examples, but you should mention shortly why and how did you proceed.

What I miss in section5: some references. You make clear and important statements here.  E.g., motion sickness or simulator sickness is a widely known problem, and has extensive literature. The same for "hallucination". The first sentence in 5.1.2. surely needs references. And also the last one about "drugs".  This is true for all other subsections here, the most important, relevant and some representative references are necessary for the reader.

In section 5 only real VR problems are highlighted. It is still confusing with your keywords AR and MR, that are totally different. In AR, you see thru glasses, generally no motion sickness is present. You handled VR scenarios throughout the paper, and no AR, however, you used this as a keyword? Please clarify.

Author Response

See word file.

Reviewer 2 Report

This is an interesting bibliometric analysis for the use of virtual reality in rehabilitation medicine. The topic is very interesting, and the article is well written.

I suggest only few aspects to improve the manuscript:

-        Pag 4, line 114. The authors explain the concept of centrality of the node. Centrality is defined with at least 3 parameters (degree centrality, closeness centrality and betweenness centrality). The authors could evaluate this topic considering centrality’s subcategories.  

-        Pag 5, line 141-142 this phrase should be deleted since it is a repetition.

Author Response

See word file.

Reviewer 3 Report

To sort out the research hotspots of the application of VR in rehabilitation medicine, analyse its research themes and research trends, and offer a reference for future related research in this field.

The authors  provide an in-depth analysis of the development process, research hotspots and research trends in the field of VR application in rehabilitation medicine research, taking the Web of Science core dataset as the source and using bibliometric analysis by CiteSpace.

Their application of VR in rehabilitation medicine was composed of three stages, and the research topics were reviewed in five aspects: neurological rehabilitation, psychology treatment, pain distraction, cardiopulmonary rehabilitation, and visual spatial disorder.

The authors concluded that:- the overcoming VR technology-induced vertigo and mental disorders from overuse of VR are both challenges to research on the application of VR in rehabilitation medicine. -In addition, developing VR products with better experience, developing standardized guidelines, and conducting more high-quality clinical studies are all future research trends for the application of VR in rehabilitation medicine.

The manuscript is interesting.

I have some minor suggestions for the authors.

1.      The abstract must better proportionally summarize the sections.

2.      The methodology described in section 2 “study design” needs to be expanded and better described.

3.      The data analysis is structured into themes and therefore into paragraphs. I suggest to add a few lines to introduce the themes.

4.      Figure 4 needs improvements.

Author Response

See word file.

Reviewer 4 Report

Dear author,

I have the following comments regarding your article:

  1. 1) In section 2.1, the author mentions using topics related to virtual reality (VR), augmented reality (AR), mixed reality (MR), and computer-mediated reality. However, based on my understanding, the title of this article seems to specifically focus on VR. Could you please clarify why the authors decided to include AR and MR in this context?

  2.  
  3. 2) In section 2.1, why not consider including databases from IEEE and Scopus together? It would be beneficial to provide a rationale for this decision.

  4.  
  5. 3) In section 3.4, I would like to understand how and why you divided the development process into three stages: Exploration Stage, Development Stage, and Promotion and Application Stage.

  6.  
  7. 4) In section 4.0, it would be helpful to include a Figure that provides an overall view of the analysis of research topics on the application of VR in rehabilitation medicine. For example, you could create a figure illustrating the topics covered in sections 4.1, 4.2, 4.3, 4.4, and 4.5.

  8.  
  9. 5) I recommend expanding the discussion on current challenges. Currently, the article only mentions two challenges, but it would be beneficial to include more than just these two. Please consider adding additional challenges to provide a more comprehensive analysis.

  10.  
  11. 6) Overall, I feel that there is a lack of explanation regarding how the author manually filtered the total number of received papers. In a systematic review, it is customary to manually remove papers that are not relevant to the topic based on predefined keywords. Please consider providing more details on this aspect.

Author Response

See word file.

Round 2

Reviewer 4 Report

Dear author:
1) Some errors in Figure 1.
2) Ready to publish. 

Author Response

See word file.
